# Pregnancy and Caffeine Metabolism: Updated Insights and Implications for Maternal–Fetal Health

**DOI:** 10.3390/nu17193173

**Published:** 2025-10-08

**Authors:** Katarzyna Maria Struniewicz, Magdalena Maria Ptaszek, Alicja Marianna Ziółkowska, Aneta Nitsch-Osuch, Aleksandra Kozłowska

**Affiliations:** 1Student Scientific Society of Hygiene and Preventive Medicine, Medical University of Warsaw, 02-106 Warsaw, Poland; s088454@student.wum.edu.pl (K.M.S.); s088396@student.wum.edu.pl (M.M.P.); s088549@student.wum.edu.pl (A.M.Z.); 2Department of Social Medicine and Public Health, Medical University of Warsaw, 02-106 Warsaw, Poland; aneta.nitsch-osuch@wum.edu.pl

**Keywords:** caffeine, pregnancy, GDM, preeclampsia, preterm birth

## Abstract

Caffeine is one of the most widely consumed psychoactive substances globally and is a common component of daily diets, particularly among women of reproductive age. Numerous in vitro and in vivo studies have indicated potential adverse effects of prenatal caffeine exposure, including disturbances in fetal growth, metabolic dysregulation, organ malformations, and neurodevelopmental alterations. These findings suggest that caffeine may influence multiple physiological pathways during gestation, including epigenetic modifications and metabolic programming. However, evidence from human studies remains heterogeneous and often inconclusive. Recent cohort studies and meta-analyses have reported that moderate maternal caffeine intake is not significantly associated with increased risks of gestational diabetes mellitus, gestational hypertension, or preeclampsia, although higher intake levels have been linked to anemia, preterm birth, and low birth weight in some populations. Furthermore, emerging data suggest potential associations between prenatal caffeine exposure and early neurodevelopmental outcomes, including behavioral changes, subtle structural brain differences, and alterations in offspring metabolic health and obesity risk. Despite these findings, the magnitude and clinical relevance of these effects remain uncertain, partly due to variability in caffeine sources, dosages, study designs, and reliance on self-reported intake. This review aims to synthesize current evidence on maternal caffeine consumption, its impact on pregnancy complications, fetal development, and long-term child health outcomes. By integrating experimental and clinical data, the study provides a comprehensive overview that may assist clinicians and healthcare professionals in counseling pregnant women regarding caffeine intake and potential risks.

## 1. Introduction

Caffeine (1,3,7-trimethylxanthine) is one of the most widely consumed methylxanthines. It occurs naturally in over 60 plant species, including coffee beans, tea leaves, cocoa beans, and cola nuts [1]. Due to its central nervous system-stimulating properties, caffeine is the most commonly used psychoactive substance worldwide, valued for its ability to enhance alertness, increase energy levels, and improve mood, thereby supporting cognitive and physical performance in daily activities. Up to 85% of the global population admits to consuming caffeine daily, with pregnant women among them [2,3,4,5].

Given its global prevalence and potential impact on pregnancy and fetal development, caffeine has been widely investigated for its biological and developmental effects. Caffeine and its metabolites are known to cross the placenta, resulting in increased catecholamine levels, which may negatively impact fetal development [2,4,6]. The literature suggests a connection between maternal caffeine consumption and increased risk of small for gestational age, preterm birth, and reduced birthweight [7,8,9,10,11]. Importantly, the effects of caffeine are not limited to the fetus but also impact the mother. Pregnancy is associated with numerous physiological changes that affect the metabolism of endogenous and exogenous substances, including caffeine. Its metabolism exhibits greater stability before pregnancy, characterized by strong correlations among caffeine metabolite levels. However, during pregnancy, there is a notable slowdown in caffeine metabolism, which results in disruptions to these correlations. Additionally, metabolism continues to slow down across trimesters, making pregnant women more susceptible to the side effects of caffeine, even if their daily intake remains the same [12,13,14].

For these reasons, pregnant women should restrict their caffeine consumption. There are no universally established guidelines regarding a safe caffeine dosage during pregnancy. The World Health Organization recommends limiting daily caffeine intake to less than 300 mg, while the American College of Obstetricians and Gynecologists advises no more than 200 mg per day [4]. Since caffeine is present in a wide range of dietary sources, it is essential for expectant mothers to monitor their intake carefully. However, differing recommendations can be confusing, potentially leading to inconsistent adherence to guidelines and unintentional overconsumption.

The issue of caffeine consumption during pregnancy has gained increasing scientific attention in recent years. A query performed on the ClinicalTrials.gov database in August 2025 using the keywords “caffeine” and “pregnancy” resulted in 71 clinical studies in which that compound was investigated. The objective of the present investigation is to clarify the correlation between the effects of caffeine consumption on pregnant patients and their offspring, regardless of whether it is via food, beverages, or dietary supplements.

## 2. Materials and Methods

We searched the Medline database via PubMed for clinical studies evaluating the effects of caffeine metabolism during pregnancy and its implications for maternal–fetal health. We used the following query:

(caffeine [Title/Abstract] OR coffee [Title/Abstract] OR “caffeine metabolism” [Title/Abstract] OR “caffeine clearance” [Title/Abstract] OR “caffeine intake” [Title/Abstract]) AND (pregnancy [Title/Abstract] OR pregnant [Title/Abstract] OR gestation [Title/Abstract] OR maternal [Title/Abstract] OR fetal [Title/Abstract] OR placenta [Title/Abstract]) AND (outcomes [Title/Abstract] OR “pregnancy complications” [Title/Abstract] OR miscarriage [Title/Abstract] OR “recurrent pregnancy loss” [Title/Abstract] OR “preterm birth” [Title/Abstract] OR “low birth weight” [Title/Abstract] OR “congenital anomalies” [Title/Abstract]) OR “gestational diabetes mellitus” [Title/Abstract]) OR “preeclampsia” [Title/Abstract]) OR “anemia” [Title/Abstract]) OR “labor dystocia” [Title/Abstract]).

Using filters, we limited article types to clinical studies, meta-analyses, or randomized controlled trials, and the publication date range from 1 January 2020 to 10 June 2025. As a result, we retrieved twenty-nine clinical papers for full-text screening.

We did not apply any publication time restrictions for preclinical data. However, we prioritized the inclusion of the most recent experimental and observational studies to provide an updated overview for readers.

## 3. Caffeine: Bioavailability and Metabolism

Caffeine (C_8_H_10_N_4_O_2_) is a purine alkaloid composed of a pyrimidine and an imidazole ring. Each molecule contains an amide, an amine, and an alkene functional group [3]. Caffeine is methylated in three positions, which makes it a 1,3,7-trimethylxanthine, an effective psychomotor stimulant [3,15].

Following ingestion, approximately 99% of this substance is absorbed into the bloodstream within 30 to 45 min [1,2]. The maximum plasma concentration occurs approximately 60 min after caffeine consumption. Concentrations increase in a dose-dependent manner due to linear pharmacokinetic characteristics. The time the peak level is reached depends on the type of dietary source. Caffeine contained in coffee is known to be absorbed faster than in carbonated drinks [2]. Factors such as age, sex, health status, and route of administration have minimal influence on caffeine’s bioavailability, which remains consistently high. This condition is primarily due to caffeine’s negligible first-pass metabolism in the liver [1,2].

In the liver, 1,3,7-trimethylxanthine is metabolized by the microsomal system of enzymes known as cytochrome P450 to other xanthines [2,16]. The cytochrome P450 1A2 isoenzyme is responsible for approximately 95% of its biotransformation [16]. These compounds, similarly to caffeine, are antagonists at adenosine receptors, with their most crucial pharmacological action being central nervous system stimulation [15,17]. Approximately 80% of caffeine is primarily demethylated to paraxanthine (1,7-dimethylxanthine), which is involved in the process of lipolysis. Approximately 11% is metabolized to theobromine (3,7-dimethylxanthine), which acts as a vasodilator and evokes mild diuresis. The remaining 4% is converted to theophylline (1,3-dimethylxanthine), a potent bronchodilator used to treat severe asthma [2,17,18]. 

In healthy adults, the average half-life of caffeine is approximately 4 to 5 h [2,16]. However, metabolic rates vary interindividually due to genetic and physiological differences. During pregnancy, CYP1A2 activity decreases, resulting in a prolonged caffeine half-life and elevated serum concentrations [13]. These changes become more pronounced as pregnancy advances, with half-life extending up to approximately 15 h in the late gestational trimester [1,13].

## 4. Dietary Sources of Caffeine and Recommendations for Pregnant and Lactating Women

As previously indicated, the half-life of caffeine varies throughout pregnancy and may extend by up to threefold. Consequently, it is recommended that pregnant and lactating women limit their caffeine intake. This topic is debated in numerous guidelines, web pages, and research studies. Although minor differences exist among them, it is pretty evident that they all concur that pregnant or lactating women should consume less caffeine than non-pregnant adults. According to the Government of Canada, the European Food Safety Authority (EFSA), and the Food Standards Agency (FSA), the recommended limit for non-pregnant adults is up to 400 mg per day. Conversely, some sources suggest different limits [19,20,21]. The World Health Organization recommends that pregnant women consider reducing their caffeine intake to below 300 mg per day [22]. This aligns with the guidance provided by the Government of Canada, which also states that pregnant or breastfeeding individuals may consume up to 300 mg of caffeine per day. However, the European Food Safety Authority has determined that the safety of the fetus is not heightened by the habitual consumption of caffeine at a dose of up to 200 mg [18]. The Food Standards Agency concurs with the point mentioned above [13]. Although recommended caffeine intake limits for pregnant and lactating women vary slightly across different guidelines, they generally converge within a range of 200 to 300 mg per day. Caffeine, as a substance found in various food sources, must be highlighted for pregnant and breastfeeding individuals to enable them to make informed and safe choices [23,24]. Table 1 provides an overview of selected beverages and products containing caffeine (Table 1).

## 5. Experimental Evidence on the Mechanisms of Caffeine Action During Pregnancy

Over the past few years, there has been a significant increase in interest regarding the impact of pregnancy on the development and functioning of offspring. In Table 2, we present studies conducted over the past 5 years using animal models. The results indicate a broad range of adverse effects on offspring associated with caffeine consumption during pregnancy, including disturbances in intrauterine growth, metabolic dysregulation, genotoxicity, organ and bone malformations, and dysfunctions across various body systems.

### 5.1. Caffeine and Adverse Developmental Outcomes in Offspring

Prenatal caffeine exposure (PCE) has been consistently linked to intrauterine growth retardation (IUGR) and impaired postnatal development in animal models. IUGR, characterized by delayed fetal growth, organ dysfunction, and reduced birth weight, increases the risk of chronic diseases in later life [26,27]. In Wistar rats, maternal caffeine administration (30 or 120 mg/kg/day, gestational days 9–20) reduced placental P-glycoprotein expression through activation of the c-Jun N-terminal kinase (JNK) pathway, resulting in elevated corticosterone levels in placental, maternal, and fetal serum. Interestingly, these changes were associated with decreased birth weight and increased incidence of IUGR (female: 66% vs. 17%; male: 69% vs. 15%) [26]. Similar findings were reported in a subsequent study using 120 mg of caffeine/kg/day, which resulted in reduced fetal weight (female: −0.99 g; male: −0.71 g) and elevated IUGR rates (female: 39% vs. 6%; male: 53% vs. 4%). Notably, both sexes exhibited postnatal catch-up growth from weeks 2 to 12 compared to the control group [27]. Consistent findings confirmed dose-dependent effects, with reduced fetal body weight and increased IUGR at 60 and 120 mg/kg/day of caffeine [28].

Postnatal growth deficits in offspring of caffeine-treated animals have been analyzed across multiple species. In Swiss mice, high-dose caffeine (1 mg/mL) resulted in lower offspring body weight at 14, 21, and 30 days compared with low-dose (0.3 mg/mL) and control groups [29]. Similarly, in Sprague–Dawley rats, maternal caffeine exposure (40 mg/kg/day, from gestational days 3.5 to 19.5) resulted in a significant reduction in fetal body weight, fetal stem length, and fetal tail length [30]. However, there is evidence that this effect may vary across species. A randomized trial in piglets found that caffeine supplementation increased neonatal weight, regardless of vitality classification [31].

PCE has also been associated with structural and functional liver alterations in rodent offspring. Reported effects include elevated glucocorticoid levels, increased expression of glucocorticoid-related genes (GR/C/EBPα), reduced fetal Insulin-like Growth Factor 1 (IGF-1), suppression of the IGF-1/IGF-1R/Akt2 pathway, and postnatal catch-up liver growth at birth [28]. Further, in female offspring, GR-/EBPα-SIRT1 pathway activation increased hepatic triglyceride accumulation, predisposing to non-alcoholic fatty liver disease (NAFLD) [32]. In male offspring, PCE in utero elevated corticosterone, inhibited autophagic flux, blocked β-oxidation, and promoted hepatic lipid accumulation, contributing to NAFLD pathogenesis [33].

Regarding studies on the impact of caffeine intake during pregnancy on bone and joint development, the harmful effects of such exposure were demonstrated. Adverse skeletal outcomes included: (i) H-type blood vessel–related long bone dysplasia via microRNA-375–mediated suppression of Connective Tissue Growth Factor (CTGF); (ii) bone growth retardation and adult osteopenia linked to maternal glucocorticoid excess and angiotensin-converting enzyme (ACE) gene hypomethylation, and (iii) downregulation of acetylation of histone H3 at lysine 9 (H3K9ac) and 11β-hydroxysteroid dehydrogenase type 2 (11β-HSD2) expression via glucocorticoid receptor/histone deacetylase 11 (GR/HDAC11) signaling, increasing osteoporosis susceptibility in adulthood [34,35,36]. Additionally, multigenerational articular damage has been linked to reduced acetylation of histone H3 at lysine 9 (H3K9ac), which affects transforming growth factor beta (TGFβ) signaling. Disruption of the glucocorticoid (GC)–insulin-like growth factor 1 (IGF1)–glucose transporter 1 (GLUT1) axis has been associated with articular cartilage dysplasia, advanced glycation end products (AGEs) accumulation, and cartilage matrix degradation, thereby increasing the risk of osteoarthritis [37,38]. Hence, compounds that target key prenatal regulatory processes can affect skeletal development, integrity, and long-term musculoskeletal health.

Neurological alterations have also been documented. In Swiss mice, PCE induced oxidative stress and genotoxicity in the brain, altered hippocampal expression of apurinic/apyrimidinic endonuclease 1 (Ape-1), BCL2-associated X protein (BAX), and B-cell lymphoma 2 (Bcl-2), and caused behavioral changes including memory deficits, depression, and anxiety [39]. In seizure models, caffeine-exposed Wistar rats exhibited a shorter latency to the first generalized seizure, whereas Genetic Absence Epilepsy Rats from Strasbourg (GAERS) displayed a delayed onset; these differences are attributed to cellular oncogene Fos (c-Fos)–mediated neuroprotective mechanisms [40].

### 5.2. Caffeine and Genotoxicity in Offspring

Genotoxicity of caffeine was evaluated in the blood, liver, and kidney tissues of offspring from mice treated with either 0.3 mg/mL or 1.0 mg/mL caffeine during copulation, pregnancy, and lactation. The studied models were assessed using the comet assay, which revealed significant Deoxyribonucleic Acid (DNA) damage in the blood, liver, and kidney tissues at 60 days of age in offspring of both sexes. The mechanism behind these impairments may involve caffeine’s effect on DNA repair in the offspring through the inactivation of cAMP Response Element-Binding Protein (CREB) [29].

### 5.3. Effects of Prenatal Caffeine Exposure on Reproductive Development

Given the negative impact of caffeine consumption on the reproductive system in postnatal life, research was conducted to investigate how prenatal exposure affects male and female fertility. Yadegari Dehnavi et al. examined the effects of embryonic caffeine exposure on ovarian follicle development. Female Wistar rats were administered 26, 45, 100, and 150 mg/kg of caffeine during pregnancy. The study concluded that high doses of caffeine caused a significant reduction in the weight, volume, and number of follicles, as well as a decrease in the thickness of the zona pellucida [41].

To explore whether abnormalities occur in male germ cells, male rat fetuses whose mothers were exposed to 30 and 120 mg/kg/day of caffeine from gestational days 9 to 20 were examined. Exposure to a high caffeine dose (120 mg/kg/day) resulted in decreased fetal body weight, disrupted testosterone production, and increased corticosterone levels. Furthermore, the expression of the Igf-1 gene in the testes decreased, and histone acetylation levels were reduced, negatively affecting testicular development. In contrast, the low caffeine dose (30 mg/kg/day) resulted in increased production of steroidogenic enzymes [42].

### 5.4. Caffeine and Offspring Metabolic Alterations

G. Chen et al. described metabolic changes in the offspring of Wistar rats. Pregnant rodents were administered 120 mg/kg of caffeine daily via intragastric administration from gestational days 9 to 20. Alterations in the metabolism of phospholipids, platelet-activating factor (PAF), arachidonic acid, bile acids, sphingosine-1-phosphate, indoxyl sulfuric acid, and cortexolone were observed. These abnormalities may potentially harm organ formation and contribute to the development of chronic diseases in adulthood in offspring with IUGR [27].

However, not all results indicated an unfavorable impact of prenatal caffeine exposure on metabolic processes. In a study by J.A. Sánchez-Salcedo et al., pregnant sows were subcutaneously administered caffeine, and their offspring were examined after birth. It was found that caffeine-treated piglets had higher partial pressure of oxygen (pO_2_) values compared to the control group (19.10 ± 0.82 vs. 14.49 ± 1.42). This may be related to increased sensitivity to carbon dioxide (CO_2_) in the respiratory center, resulting from the antagonistic actions of caffeine at the Adenosine A1 and Adenosine A2A receptors. This effect may enhance the vitality scores of piglets, which was also observed in the study (8.72 ± 0.12 compared to 7.28 ± 0.16). Interestingly, other metabolic values were comparable between the groups [31].

### 5.5. Caffeine and Disorders of the Respiratory and Cardiovascular Systems

Evidence has shown that a harmful intrauterine environment can increase the risk of cardiovascular diseases later in life. Prenatal exposure to caffeine may disrupt vascular function in the cerebral arteries of aged individuals. This alteration is associated with dysfunction in the Protein Kinase A/Ryanodine Receptor/Big Potassium Channel, Calcium-activated Potassium Channel (PKA/RyR/BKCa) signaling pathway, which plays a critical role in regulating vascular function. Such changes impair the ability of isoprenaline (a beta-adrenergic agonist) to dilate the middle cerebral artery in rat offspring, possibly increasing the risk of arterial vasospasm or stroke. However, it was noted that this study did not include any epigenetic analyses, and the influence of the endothelium on changes in vascular function, along with the role of beta-adrenergic receptors, should be considered [30].

In the study conducted by Gwon et al., a single administration of caffeine to mouse embryos did not cause significant damage to embryonic development compared to the control group. However, co-exposure to caffeine and bisphenol A (a chemical used in the production of cans) resulted in considerable cardiovascular anomalies during offspring maturation. These issues included a reduced thickness of the ventricular wall, trabecular heart disorders, impaired vasculogenesis, decreased expression of vasculogenic growth factors, and apoptotic cell death. The cause of these problems was linked to changes in the messenger ribonucleic acid (mRNA) levels of anti-oxidative, apoptotic, and hypoxic genes [43].

Recent studies also examined effects on the respiratory system, specifically asthma susceptibility in mice exposed to caffeine prenatally. Pregnant mice were administered 96 mg/kg of caffeine daily from gestational days 9 to 18, and their offspring were then subjected to ovalbumin sensitization and challenge. PCE reduced CD4+ T thymopoiesis by enhancing thymus autophagy, which caused suppression and alteration of pulmonary CD4+ T-cells, increasing allergic asthma susceptibility. The mechanism behind this phenomenon appears to involve caffeine blocking Adenosine A2A receptor/Protein Kinase A (A2AR–PKA) signaling in thymocytes and upregulating the Beclin1–LC3II autophagy pathway, leading to the degradation of p62–B-cell lymphoma 10 (p62–Bcl10), decreased A20 protein expression, and ultimately impairing the survival of CD4+ single-positive (CD4+SP) cells [44].

**Table 2 nutrients-17-03173-t002:** Recent in vivo or in vitro studies on prenatal caffeine exposure and offspring metabolic outcomes in experimental models.

Disorder/Substances	In Vitro or In Vivo Model		Mode of Action	References
Intrauterine Growth Restriction				
Caffeine (120 mg/kg × day) separately or combined with sodium ferulate (50 mg/kg × day)	Pregnant mice from GD 9 to GD 18	↓	histone acetylation	Ge et al. [26]
			inhibition of P-gp expression via active- tion of RYR/JNK/YB-1/P300 pathway	
Caffeine (30 mg/kg × day (low dose) and 120 mg/kg × day (high dose))	Pregnant Wistar rats from GD 9 to GD 20	↓	histone acetylation	
			inhibition of P-gp expression via activetion of RYR/JNK/YB-1/P300 pathway	
Caffeine (0, 0.1, 1, 10, and 10 μM) for 48 h	Primary human trophoblasts and BeWo cells supplemented with 10% fetal bovine serum and 0.1% penicillin/streptomycin at 37 °C in a 5% CO_2_ humidified incubator	↑	RYR1 and RYR3 mRNA expression levels	
Caffeine (120 mg/kg × day)	Pregnant Wistar rats from GD9 to GD20	↓	fetal bodyweights	Chen et al. [27]
			catch-up growth after birth	
Liver Developmental Dysfunctions and Diseases in Offspring				
Caffeine (30, 60, and 120 mg/kg × day)	Pregnant Wistar rats		inhibition of GR/C/EBPα/IGF-1R signaling	He et al. [28]
Caffeine (30, 60, and 120 mg/kg × day)	Pregnant Wistar rats from GD 9 to GD 20	↑	expression levels of hepatic GR,	Hu et al. [32]
		↓	C/EBPα, FASN, and SREBP1c	
		↑	SIRT1 expression	
			levels of H3K14ac and H3K27ac in the SREBP1c and FASN gene promoters	
Caffeine (60 and 120 mg/kg × day)	Pregnant Wistar rats from GD 9 to GD 20		β-oxidation inhibition and lipid accumulation in the liver caused by inhibition of cathepsin D expression in hepatocytes lysosomal degradation dysfunction and autophagy flux blockade caused by upregulation of miR-665	Zhang et al. [33]
Bone and Articular Abnormalities in Offspring				
Caffeine (30, 60, and 120 mg/kg × day)	Pregnant rats from GD 9 to 20	↓	expression of CTGF by miR375	He et al. [34]
			FAK inhibition and adverse H-type blood vessel formation	
		↓	mRNA expression of Col 1, Osterix, RUNX2, and Osteocalcin genes	
Caffeine (120 mg/kg × day)	Pregnant rats from GD 9 to 20		H3K9ac and expression levels of 11β-HSD2	Xiao et al. [36]
			promotion of GR in BMSCs	
		↓	recruitment of HDAC11	
Caffeine (120 mg/kg × day)	Pregnant rats from GD 9 to 20	↓	H3K9 acetylation and expression of the TGFβ signaling pathway	Zhao et al. [37]
Caffeine (12 mg/100 g × day)	Pregnant rats on GD from 9 to 20, BMSCs were treated with exogenous corticosterone during osteogenic induction	↑	GR and ACE gene expression	Wen et al. [35]
		↓	BGLAP, ALP, and BSP gene expression	
		↑	hypomethylation of the ACE promoter	
			angiotensin 2 content	
Caffeine (120mg/kg × day)	Pregnant rats from GD 9 to 20		mediation of intrauterine dysplasia of articular cartilage by the GC-IGF1-GLUT1 axis mediation of increased accumulation of AGEs and matrix degradation by the GC-IGF-1-GLUT1 axis	Qing-Xian et al. [38]
Changes in Sex Glands in Offspring				
Caffeine (26, 45, 100, and 150 mg/kg doses of caffeine via drinking water)	Pregnant rats	↓	total volume of the ovaries	Yadegari Dehnav et al. [41]
		↓	number of primary and secondary follicles	
		↓	diameter of ovarian follicles	
		↓	volume of the oocyte	
		↓	zona pellucida thickness	
Caffeine (30 and 120mg/kg × day)	Pregnant Wistar rats from GD 9 to GD 20	↓	testicular IGF-1 expression	Pei et al. [42]
		↓	H3K14ac level in the IGF-1 promoter region	
			body weight	
		↓	inhibition of testosterone synthetic function	
			30 mg/kg × day:	
		↑	steroidogenic enzyme expression	
Genotoxicity in Offspring				
Caffeine (0.3 or 1.0 mg/mL) during copulation (7 days), pregnancy (21 days), and lactation (21 days)	Swiss female mice (60 days old)		DNA repair deficiency via CREB inactivation	Lummertz Magenis et al. [29]
Cardiovascular Disorders in Offspring				
Caffeine (30, 60, and 120 μg/mL) and BPA (35 μg/mL) for 48 h	Mouse embryos with a yolk sac placenta		alteration of mRNA levels of anti-oxidative, apoptotic, and hypoxic genes abnormal vasculogenesis apoptotic cell death	Gwon et al. [43]
Metabolic Profile Changes in Offspring				
Caffeine (120 mg/kg × day)	Pregnant Wistar rats from GD9 to GD20	↓	serum PAF level in female fetuses	Chen et al. [27]
		↑	serum PAF level of male offspring	
		↓	levels of fetal serum phospholipids, bile acid, sphingosine-1-phosphate, and cortexolone in female offspring	
			abnormal renal function in offspring	
Caffeine (210 mg × day, at a volume of 4.2 mL, administered subcutaneously)	Multiparous (3.6 ± 0.32 births; weight: 200–280 kg) Yorkshire-Landrace sows on GD from 113 to 114		antagonistic actions of caffeine at the adenosine A_1_ and A_2A_ receptors at the respiratory center	Sánchez-Salcedo et al. [31]
Dysfunction of Aged Cerebral Arteries in Offspring				
Caffeine (20 mg/kg, 2 × day)	Pregnant Sprague-Dawley rats on GD from 3.5 to 19.5	↓	down-regulation of the PKA	Li et al. [30]
		↓	down-regulation of the RYR	
		↓	down-regulation of the large-conductance Ca^2+^-activated K^+^ pathway	
Respiratory Dysfunctions in Offspring				
Caffeine (96 mg/k × day)	Pregnant Wistar rats from GD9 to GD20	↑	levels of IL-4 and IL-17A	Liu et al. [44]
		↓	levels of IFN-γ and TNF-α	
		↑	Th2/Th1 cells ratio and Th17/Tregs cells ratio	
		↓	A2AR-PKA signaling	
		↑	Beclin1-LC3II autophagy	
		↑	Bcl10 degradation	
		↓	A20 expression	
			inhibition of CD4+ thymopoesis	
Neurodevelopmental Changes in Offspring				
Caffeine (1.0 or 0.3 mg/mL during copulation (7 days), pregnancy (21 days), and lactation (21 days)	Swiss female mice (60 days old)		changes in Ape-1, BAX, and Bcl-2 in the female offspring hippocampus at 30 days of life	Magenis et al. [39] (abstract only)
Caffeine dissolved in water (0.3g/L) before conception, during pregnancy, and during the lactation period	Wistar rats and GAERS Offspring on PN30 subjected to 70 mg/kg of PTZ	↓	c-Fos protein expression	Yavuz et al. [40]

↑—increase, ↓—decrease, 11β-HSD2—11β-hydroxysteroid dehydrogenase 2, A20—Tumor Necrosis Factor Alpha-Induced Protein 3, A2AR—A2 adenosine receptor, ACE—angiotensin-converting enzyme, AGE—advanced glycation end products, ALP—alkaline phosphatase, Ape-1—apurinic endonuclease 1, BAX—Bcl-2 Associated X-protein, Bcl10—B-cell lymphoma-10 protein, Bcl-2—B-cell lymphoma-2 protein, Beclin1–LC3II—Bcl-2 interacting cell death executor protein 1-Microtubule-associated protein 1A/1B-light chain 3, lapidated form, BGLAP—bone gamma-carboxyglutamate protein, BMSCs—bone marrow mesenchymal stem cells, BSP—bone sialoprotein, C/EBPα—CCAAT enhancer-binding protein α, Col 1—collagen type 1, CTGF—connective tissue growth factor, FAK—focal adhesion kinase, FASN—fatty acid synthetase, GAERS—genetic absence epilepsy rats from Strasbourg, GC—glucocorticoid, GD—gestational day, GLUT1—glucose transporter 1, GR—glucocorticoid receptor, H3K14ac—histone H3 acetylated 14 lysine, H3K27ac—histone H3 acetylated 27 lysine, HDAC11—histone deacetylase 11, IFN-γ—interferon gamma, IGF-1—Insulin-like Growth Factor 1, IL—interleukin, PAF—Platelet-Activating Factor, PKA—Protein Kinase A, PTZ—pentylenetetrazole, RYR—ryanodine receptor, SREBP1c—Sterol regulatory element-binding protein 1c, TGFβ—transforming growth factor beta, TNF-α—tumor necrosis factor alpha.

## 6. Clinical Studies on Caffeine Exposure During Pregnancy—An Update from the Last 5 Years

### 6.1. Impact of Caffeine Consumption on Maternal Pregnancy Complications

Numerous studies have investigated the impact of caffeine on maternal metabolism and its correlation with the development of gestational diabetes mellitus (GDM) as well as preeclampsia, gestational hypertension (GH), and pregnancy anemia [45,46,47]. This area of research underscores the importance of understanding how caffeine digestion impacts the course of pregnancy (Appendix A).

#### 6.1.1. Maternal Caffeine Intake and Risk of Gestational Diabetes Mellitus

GDM constitutes a clinically relevant metabolic disorder associated with considerable short- and long-term risks for both maternal and fetal health [48]. The potential influence of caffeine consumption on the development of GDM has been the subject of several recent investigations. In a study by Kukkonen et al., participants were stratified according to both the amount and source of caffeine intake. The results demonstrated that moderate coffee consumption during the first trimester was not associated with an increased risk of GDM. Interestingly, age-adjusted models suggested a potential protective effect. By contrast, despite the relatively low prevalence of cola consumption within the cohort, higher intake levels were positively correlated with an elevated risk of GDM [47]. These findings are consistent with those of Hinkle et al., who examined caffeine intake in relation to GDM in a prospective cohort study [45]. The analysis revealed no significant association between caffeine consumption during the 10- to 13-week period of gestation and GDM risk. However, women consuming up to 100 mg/day of caffeine during 16 to 22 weeks exhibited a 47% reduction in the likelihood of developing GDM compared with women reporting no caffeine intake. Similarly, an inverse association was observed for intakes up to 200 mg/day of caffeinated beverages. Moreover, the study assessed plasma concentrations of caffeine and its primary metabolite, paraxanthine, and these biomarkers were not significantly associated with either GDM incidence or impaired glucose tolerance (IGT). Nonetheless, women in the highest quartile of total caffeine and paraxanthine levels demonstrated mean glucose concentrations 3.8 mg/dL lower than those in the lowest quartile.

#### 6.1.2. Maternal Caffeine Intake and Risk of Hypertensive Disorders of Pregnancy

Given the established role of hypertensive disorders as major contributors to maternal and perinatal morbidity, several studies have examined whether caffeine intake during pregnancy influences the risk of gestational hypertension and preeclampsia [48,49]. In a cohort study, the associations between caffeine consumption and the subsequent risks of GH and preeclampsia were investigated. The results indicated no correlation between the consumption of caffeinated beverages and the occurrence of these conditions, nor was there a significant relationship between plasma levels of caffeine or paraxanthine and gestational complications. Furthermore, no notable differences in blood pressure were found across gestational periods in relation to beverage intake or plasma caffeine levels. Notably, intake of caffeinated beverages during the period from 16 to 22 weeks of gestation was linked to specific enhancements in the fasting cardiometabolic profile [45]. Similar results were observed in a meta-analysis of ten studies, which included 114,984 pregnant women. The analysis indicated no meaningful association between caffeine intake during pregnancy and the risks of GH or preeclampsia. Furthermore, comparisons between low to moderate caffeine consumption and no or minimal intake demonstrated that caffeine exposure during pregnancy was not significantly linked to GH or preeclampsia [15].

#### 6.1.3. Maternal Caffeine Intake and Risk of Anemia During Pregnancy

Anemia during pregnancy has been extensively studied, with multiple contributing factors identified in the literature. Reported determinants include caffeine consumption, maternal age, educational attainment, alcohol use, gestational age, family size, and overall nutritional status. Notably, pregnant women who consumed caffeine-containing beverages (coffee or tea) even on an occasional basis demonstrated a twofold higher risk of developing anemia compared with non-consumers [46]. This association may result from caffeine’s interference with non-heme iron absorption in the gastrointestinal tract, potentially decreasing maternal iron bioavailability and contributing to the development of anemia during pregnancy [50].

### 6.2. Caffeine Consumption and Adverse Pregnancy and Labor Outcomes

#### 6.2.1. Maternal Caffeine Intake and Risk of Preterm Birth

The metabolism of caffeine has been studied to investigate its potential relationship with pregnancy outcomes and labor complications, including premature labor. A case–control analysis investigated the correlation between preterm birth and 812 molecular features identified in plasma. The findings indicated variability in substance concentrations, with specific molecules exhibiting elevated levels in preterm birth cases compared to term controls. Caffeine was identified among these molecules, demonstrating a significant fold change. This indicates that its higher concentration within the preterm birth cohort may suggest a potential influence on the risk of preterm birth [11]. This statement aligns with the findings of a different study. A prospective birth cohort differentiated between infants born to mothers whose caffeine metabolism fell within the lowest quartile of consumption (4.2–86.4 mg/day) and those whose caffeine intake was in the highest quartile (205.5–5080 mg/day). The findings demonstrated that offspring of mothers in the highest quartile of caffeine intake exhibited a substantially greater risk of preterm birth during the second trimester (RR = 1.94, 95% CI: 1.12–3.37), corresponding to an approximately 94% increase in risk compared with the lowest intake group [7].

#### 6.2.2. Maternal Caffeine Intake and Risk of Recurrent Pregnancy Loss

Recurrent pregnancy loss (RPL) represents a significant complication in obstetrics that has received considerable attention in the literature [51]. A recent systematic review and meta-analysis explored the association between female lifestyle factors and the risk of RPL. Caffeine consumption was examined among the various factors evaluated, revealing a notable correlation across four studies. The findings suggest that the probability of RPL is higher in individuals with elevated caffeine intake compared to those with lower consumption; however, this difference did not achieve statistical significance. A closer investigation reveals a dose-dependent association, with the highest risks observed at intake levels exceeding 300 mg per day. Nevertheless, results remain inconsistent, particularly as no effect was observed among smokers, and several studies found no clear association in the general population [52].

#### 6.2.3. Maternal Caffeine Intake and Risk of Labor Dystocia

A systematic review and meta-analysis conducted by Jochumsen et al. specifically investigated maternal determinants of labor dystocia, a clinically significant obstetric complication. Only one included study reported that pregnant women consuming 200–299 mg of caffeine per day had a higher prevalence of labor dystocia compared with those consuming 0–99 mg. However, trend analysis across varying intake levels did not demonstrate a significant overall association between caffeine consumption and labor dystocia [53]. 

### 6.3. Epigenetic and Microbiome Effects of Prenatal Caffeine Exposure

#### 6.3.1. Epigenetic Effects of Prenatal Caffeine Exposure

A meta-analysis of six international cohorts investigated the influence of maternal caffeine metabolism during pregnancy on DNA methylation (DNAm) in cord blood [54]. The analysis identified two CpG sites significantly associated with maternal caffeine intake. The cg19370043 site was related to total caffeine consumption, predominantly driven by coffee intake, whereas the cg14591243 site was linked explicitly to caffeine derived from cola beverages. At the latter locus, each additional cup of cola was associated with a 0.06% increase in DNAm. In total, 12–22 differentially methylated regions (DMRs) were detected in relation to caffeine exposure, with the most prominent region located on chromosome 17 and observed across coffee, tea, and cola consumption. Sex-stratified analyses further revealed differences in DMR associations between male and female offspring. Overall, these findings highlight the modest, beverage-specific effects of maternal caffeine intake on neonatal DNA methylation, suggesting that additional environmental or biological factors beyond caffeine may contribute to the observed epigenetic patterns. A similar study examined the relationship between DNA methylation and caffeine metabolites in cord blood, with a particular focus on theobromine levels at 8 weeks of gestation. This investigation revealed only subtle alterations and did not provide evidence for widespread epigenetic effects [14].

#### 6.3.2. Prenatal Caffeine Exposure and Offspring Gut Microbiome

Beyond epigenetic modifications, another line of research has explored the potential impact of prenatal caffeine exposure on the gut microbiome. A prospective cohort study examined the effect of prenatal exposure to caffeine and acetaminophen on the gut microbiome. The results indicated that prenatal exposure to caffeine did not show a significant association with alterations in alpha diversity or the relative abundance of bacterial species, phyla, or functional pathways within the gut microbiome. Despite the absence of substantial changes in the gut microbiome correlated with caffeine exposure, specific gene pathways, particularly those associated with methionine synthesis, demonstrated an increased relative abundance. These findings suggest that the observed effect may be attributable to caffeine exposure, which appears to enhance the presence of methyl groups, as the metabolism of caffeine involves processes of demethylation [55].

### 6.4. Impact of Prenatal Caffeine Exposure on Child Development and Health

#### 6.4.1. Effects on Fetal Growth and Birth Outcomes

Caffeine’s influence on children’s development has been widely investigated in studies over the last 5 years. Children born to mothers who consumed caffeine during pregnancy manifested reduced fetal development measures, as well as delays in motor skills [7,8,9,10,56].

In an extensive prospective study, children whose mothers consumed more than 300 mg of caffeine per day during pregnancy exhibited developmental delays at 6 and 12 months old compared to the reference group (<100 mg caffeine/day). Interestingly, differences in measurements were observed between the 6- and 12-month assessments. Children aged 6 months presented delays in communication, fine motor, problem-solving, and personal-social skills. During the 12-month evaluation, the most significant delays were observed in motor skills [56]. Therefore, there is a possibility that high maternal caffeine intake during pregnancy contributes to early developmental delays. 

Several studies have identified a dose-dependent correlation between caffeine intake and delays in fetal development. Increasing caffeine consumption harmed the anthropometric measures of neonates. Women who reported drinking >50 mg of caffeine daily gave birth to children with lower birth weight (LBW) and length, reduced arm and thigh circumference, and smaller anterior thigh skin fold compared to no intake [10]. In a different study, children of women who consumed caffeine during pregnancy had reduced height z-scores compared to those with no exposure [9]. These findings are consistent with two more studies, in which daily caffeine intake below the recommended dosage (<200 mg) was associated with reduced birth weight and a small for gestational age (SGA) [7,8]. The dose-dependent risk of LBW has also been shown in meta-analyses. Each additional 100 mg of caffeine a day increased the risk of LBW [57,58]. Conversely, a meta-analysis by Brito Nunes et al. identified an association between maternal coffee consumption and higher birth weight, yet the findings were inconsistent [59].

#### 6.4.2. Caffeine and Early Neurodevelopmental Outcomes

Several studies have investigated the potential association between intrauterine caffeine exposure and the subsequent risk of attention-deficit hyperactivity disorder (ADHD) as well as congenital malformations in offspring [60,61,62]. Haan et al. found no evidence supporting a relationship between caffeine consumption during pregnancy and an increased risk of offspring ADHD symptoms [60,62]. Regarding birth defects, a total caffeine consumption of ≥300 mg/day was modestly associated with craniosynostosis and aortic stenosis in children. Lower maternal caffeine intake (10 to 300 mg/day) was associated with small, but statistically significant increases in adjusted odds ratios (aORs) for 10 of the 48 studied defects. Interestingly, the most important associations were found for caffeinated soda consumption rather than other caffeinated beverages, suggesting potential confounding by the presence of sweeteners [61].

Regarding the effect of caffeine on neurodevelopment, prenatal exposure was associated with changes in white matter and higher psychopathology measures in children aged from 9 to 11 [63,64]. An analysis of the Adolescent Brain Cognitive Development Study (ABCD Study), which included over 9000 participants, revealed less favorable outcomes in externalizing and internalizing behaviors, somatization, and neurodevelopmental measures in children exposed to caffeine prenatally [63]. A subsequent analysis of the ABDC dataset revealed an association between caffeine consumption and difficulties with externalizing behavior, but not with internalizing behavior. What is more, the recommended dose of caffeine during pregnancy increased the likelihood of more severe somatic, oppositional defiant, and conduct issues [64]. Both studies remain consistent that gestational caffeine exposure has no impact on cognitive difficulties in children [63,64].

#### 6.4.3. Risk of Neurological and Neurodevelopmental Disorders

A meta-analysis of Health Outcomes and Measures of the (HOME) and Early Autism Risk Longitudinal Investigation (EARLI) studies identified a moderate positive correlation between higher maternal caffeine intake and autism spectrum disorder (ASD)-related behavior in children aged from 3 to 8. The behavior was assessed using the Social Responsiveness Scale (SRS). The association was more pronounced in the EARLI Study, whereas in the HOME Study, increasing pre-pregnancy body mass index (BMI) scores in mothers were associated with higher SRS T-scores. However, we should acknowledge that the SRS test results do not necessarily equate to an ASD diagnosis [65].

#### 6.4.4. Risk of Childhood Neoplasms

While investigating the potential role of intrauterine caffeine exposure in childhood diseases, the risk of developing neoplasms was a subject of debate in a few studies. A study on children with acute myeloid leukemia (AML) provided minor evidence of an increased risk of AML in those born to mothers who consumed coffee during pregnancy. On the other hand, tea consumption was negatively correlated with hyperdiploidy (≥47 chromosomes). However, the analysis was based on a limited number of cases [66]. Similarly, Hu et al. suggest that maternal coffee, but not tea consumption, may increase the risk of childhood brain tumors (CBTs), especially glioma [67]. These findings highlight the potential role of prenatal caffeine exposure in the risk of childhood neoplasms and the need for further research [66,67].

#### 6.4.5. Impact of Prenatal Caffeine Exposure on Offspring Fertility

A study by Eubanks, A.A., et al. addressed the issue of reproductive health in women exposed to caffeine in utero. Anti-Müllerian hormone (AMH), a widely used marker of ovarian reserve, tended to be lower (<1.0 ng/mL) in the group with prenatal caffeine exposure (RR = 1.90, 95% CI: 1.09–3.30). These findings suggest that maternal caffeine intake during pregnancy may have a negative impact on offspring fertility [68].

#### 6.4.6. Metabolic Health and Obesity Risk in Children

Studies on caffeine’s effect on metabolic disorders in children indicate that higher caffeine intake may be associated with an increased BMI and the risk of childhood obesity [9,58,69,70,71]. Children born to mothers who consumed ≥180 mg of caffeine per day presented a higher total body fat index and increased liver fat fraction at the age of 10 [69]. A meta-analysis of 15 studies revealed a 0.27 (95% CI: 0.02, 0.52) change in BMI z-score between the highest and lowest caffeine intake groups. Moreover, each additional 100 mg of daily caffeine consumption was associated with a 31% higher relative risk (RR = 1.31, 95% CI: 1.11, 1.55) of obesity [58]. Furthermore, coffee as a source of caffeine may have a greater impact on children’s obesity and adiposity risk than tea [71]. However, a study investigated by Gleason et al. found no connection between maternal caffeine consumption and changes in their offspring’s BMI [9].

## 7. Conclusions

Pregnancy is a crucial period in which maternal physiology undergoes profound adaptations to support fetal development. Among numerous environmental and lifestyle factors, caffeine consumption remains a relevant exposure during pregnancy. This raises important questions regarding its safety and potential metabolic implications for both mother and fetus.

The present narrative review synthesizes current evidence on the effects of maternal caffeine intake during pregnancy. Over the past five years, numerous animal and in vitro studies have demonstrated potentially harmful effects of caffeine exposure, including intrauterine growth disturbances, metabolic dysregulation, organ malformations, and genotoxicity. However, these findings have not been consistently confirmed in human studies. Clinical investigations into gestational diabetes mellitus have shown no clear association between moderate caffeine consumption and increased risk, with some data even suggesting a protective effect at low to moderate intake levels. Similarly, no significant relationships have been observed between caffeine consumption and gestational hypertension or preeclampsia, while occasional improvements in cardiometabolic profiles have been reported. By contrast, caffeine intake has been associated with an increased risk of maternal anemia, likely due to impaired iron absorption. Evidence regarding preterm birth, recurrent pregnancy loss, and labor dystocia remains inconsistent, with some studies suggesting dose-dependent risks at higher intake levels, but others finding no significant associations. However, fetal growth studies indicated that even low-to-moderate maternal caffeine intake may be associated with reduced birth weight, small-for-gestational-age risk, and impaired neonatal anthropometrics. Overall, current data highlight discrepancies between preclinical and human research, as well as between different clinical studies.

Overall, current data highlight significant discrepancies between preclinical and human research, as well as between different clinical studies. The marked discrepancy between pronounced toxicity, such as genotoxicity and organ abnormalities, observed in animal models and the comparatively less evident manifestation in human populations at estimated standard exposure levels necessitates careful consideration. This divergence is likely attributable to significant species-specific differences in toxicokinetics, notably metabolism and elimination, which often confer a higher biological resilience or more rapid detoxification in humans. Furthermore, this discrepancy is exacerbated by the usually substantially greater relative doses employed in animal studies compared to typical human environmental exposure, which may introduce findings with limited direct translational value to standard human consumption patterns.

Our review does not provide personalized or trimester-specific recommendations based on individual metabolic clearance rates. The difficulty in formulating such precise guidelines stems from the high genetic variability of CYP1A2 activity and the lack of objective measurements of plasma caffeine clearance in current clinical practice and in most cohort studies (which rely on self-reported intake). Therefore, safety guidelines remain population-based and rely on a daily dose (mg/day), rather than on a dynamic, personalized rate of metabolism. If the half-life extends to over 15 h during pregnancy, it seems that a woman should consume caffeine less frequently than once every half-life to prevent accumulation. Therefore, instead of saying “no more than 200 mg/day,” a new, more pharmacokinetically justified recommendation could be as follows: in pregnancy, caffeine consumption should be avoided or limited to a maximum of one caffeinated beverage every 24–36 h.

Regarding caffeine’s effects on fetal–maternal health, future research should focus on establishing clear dose–response relationships to define safe thresholds of caffeine intake during pregnancy. The main limitations of the presented clinical data include variability in caffeine sources and dosages, reliance on self-reported intake, measurement errors in caffeine exposure, differences in individual caffeine metabolism rates, and limited consideration of confounding factors, such as smoking status, CYP1A2 polymorphisms, BMI, iron status, and overall dietary patterns, which may partly explain inconsistencies across studies. It should also be noted that in most of the studied populations, caffeine consumption largely remained within or only slightly above the recommended intake levels. Therefore, extrapolation of these findings to scenarios of habitual high caffeine consumption during pregnancy is restricted. A key limitation of the present review is the temporal restriction applied to the clinical search, which excluded earlier potentially pertinent, high-impact observational research, particularly concerning long-term outcomes in offspring. While this approach emphasized the most current evidence, it may have inevitably influenced the comprehensiveness of the synthesis regarding outcomes established by highly cited foundational studies before 2020, representing a trade-off between currency and historical breadth. Further large-scale, well-controlled studies are warranted to clarify the role of caffeine metabolism in pregnancy-related health outcomes.

In conclusion, the findings of this review suggest that while experimental studies consistently indicate potential risks of caffeine exposure during pregnancy, current human evidence remains inconclusive. This review may help clinicians and healthcare professionals counsel pregnant women on safe caffeine consumption, emphasizing moderation and the importance of individualized assessment. Clinical implications of these findings, along with practical guidance for clinicians and dietitians, are summarized in Table 3.

## Figures and Tables

**Table 1 nutrients-17-03173-t001:** Caffeine content in selected food and beverages, based on [25].

Category/Products	Caffeine (mg/100 g or mL)	Category/Products	Caffeine (mg/100 g or mL)
Coffes		Chocolate	
Brewed Coffee, Arábica	30	Milk chocolate	19
Powder Coffees	1165–1444	Semisweet Chocolate	70
Espresso Coffee	279	Dark Chocolate	114
Instant Coffee (soluble), powder	3344	Desserts	
Frappuccino Coffee, Starbucks	25	Coffee Pudding	22
Cappuccino Coffee	32	Coffee Cake	35
Brewed Coffee, Arábica with milk (80% coffee: 20% milk)	24	Coffee Mousse	67
Teas and Infusions		Tiramisu	9
Green Tea, infused	20	Brownie	18
Black Tea, infused	18	Drinks	
Yerba Mate	24–26	Cola Soda	9
Iced Tea	6	Energy drinks	30
Rooibos Tea (red), infused	16	Dietary Supplements	
Cocoa		Guarana, powder	3044
Cocoa, powder	230	Energy Bar/Gel with caffeine	167–233

**Table 3 nutrients-17-03173-t003:** Take-home box for clinicians and dietitians: prenatal caffeine management.

Category	Recommended Thresholds & Sources	Clinical Implications & Guidance
Indicative Daily Intake Thresholds (mg/day)	Maximum recommended intake: ≤200 mg/day (ACOG, EFSA, FSA). Some sources suggest a daily intake of ≤300 mg (WHO).	Consumption of more than 200 mg/day is consistently associated with increased risks. Metabolism slows significantly (with a half-life of up to 15 h in late pregnancy), thereby improving maternal/fetal exposure.
	Therefore, instead of saying “no more than 200 mg/day,” a new, more pharmacokinetically justified recommendation could be: In pregnancy, caffeine consumption should be avoided or limited to a maximum of one caffeinated beverage every 24–36 h.	
Associated Risks with High Intake	Intake >300 mg/day	Linked to increased risk of Small for Gestational Age (SGA), reduced birth weight, preterm birth, and anemia (especially in the third trimester). Intake >300 mg/day linked to gross motor developmental delay in children at 12 months.
Caffeine Sources (Examples & Content)	Coffee, Tea (Black/Green), Dark Chocolate, Energy Drinks, Guarana (Powder), Cocoa (Powder)	Counsel patients to count all sources, including dark chocolate, soft drinks (such as cola drinks, which are linked to an increased risk of GDM), and supplements (e.g., Guarana—extreme caution is warranted due to its high concentration).
Trimester-Specific Guidance	Restriction applies throughout pregnancy.	The risk of SGA appears more significant with high intake during the early pregnancy (first trimester), as the organs are undergoing critical development. Continuous high exposure is detrimental across all trimesters.
Timing Relative to Iron Supplementation	Avoid consumption of caffeine-containing beverages around the time of supplementation.	High caffeine intake is associated with maternal anemia. Coffee and tea are known to impair non-heme iron absorption in the gut. Advise a gap (e.g., 1–2 h) between caffeine and iron supplements to maximize iron status.

ACOG—American College of Obstetricians and Gynecologists, EFSA—European Food Safety Authority, FSA—Food Standards Agency, WHO—World Health Organization, SGA—Small for Gestational Age, GDM—Gestational Diabetes Mellitus.

## Data Availability

The data presented in this study are available on request from the corresponding author.

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
