# Peer review of "Pregnancy and Caffeine Metabolism: Updated Insights and Implications for Maternal–Fetal Health"

_nutrients, 2025, doi:10.3390/nu17193173_

Round 1

Reviewer 1 Report

Comments and Suggestions for Authors

An interesting review entitled "Pregnancy and Caffeine Metabolism: Updated Insights and Implications for Maternal–Fetal Health
This review is a comprehensive synthesis of recent experimental and clinical studies (2020–2025) and integrates molecular, epigenetic, and clinical data into a translational framework with practical relevance: it provides useful information for counselling pregnant women.
However, as with any narrative review, there is a risk of bias in the selection of studies. I suggest standardising the definition of ‘high’ or ‘moderate’ consumption across the included studies. Mention in the weaknesses in the evidence that there is a reliance on self-reporting in clinical studies, with possible underestimation of actual consumption. In addition, the format of the references, which is not standardised, should be reviewed.

Author Response

a

Reviewer 2 Report

Comments and Suggestions for Authors

Parameters and Clarification of Evidence

1. Harmonizing Preclinical and Clinical Data: The review synthesizes both experimental (preclinical) and human (clinical) information. Considering that the authors acknowledge preclinical research regularly suggests possible dangers, but human data are varied and often inconclusive, does the discussion sufficiently address this discrepancy? Should the authors hypothesize why the pronounced negative consequences shown in animal models (e.g., genotoxicity, organ abnormalities) do not more evidently manifest in human populations at standard exposure levels?

2. Addressing Research Heterogeneity: The abstract clearly states that the findings are questionable due to diversity in caffeine supplies, doses, research designs, and reliance on self-reported consumption. How does the review methodically address these confounding variables when integrating the data, namely in the included meta-analyses or cohort studies?

Clinical Significance and Dose-Response Relationship

3. The study highlights a crucial discovery: a significant slowdown in caffeine metabolism during pregnancy, with the half-life extending to over 15 hours in the later stages of gestation. What is the relationship between the synthesis of clinical outcomes and this physiological change? Does the review provide a more customized or trimester-specific recommendation for caffeine consumption based on metabolic clearance rates?

4. The document highlights the presence of varying recommendations (e.g., WHO: <300 mg/day; ACOG/EFSA: <200 mg/day), which may create confusion for patients and healthcare professionals. Does the review provide a conclusive, evidence-based suggestion for a safe threshold for caffeine intake that would be beneficial for healthcare professionals?

Approach

5. The clinical search was limited to clinical studies, meta-analyses, or randomized controlled trials published between January 1, 2020, and June 10, 2025. Was the removal of pertinent, high-impact earlier research (even if observational) on specific long-term offspring outcomes appropriate, and how may this temporal filter influence the comprehensiveness of the current review?

6. Critique of Exposure Assessment: Given that dependence on self-reported consumption is identified as a significant constraint, does the evaluation sufficiently evaluate the methodologies used to assess caffeine exposure in the human studies included? Were studies using objective biomarkers (e.g., paraxanthine levels) evaluated or addressed differently compared to those relying only on dietary questionnaires? 

Comments on the Quality of English Language

The English might be enhanced to convey the study more effectively. 

Author Response

Thank you kindly for the review. I've attached the revised response.

Reviewer 3 Report

Comments and Suggestions for Authors

Dear Authors,

Thank you for the opportunity to review the manuscript, in which you address a topic of exceptional importance to public health and clinical practice: the widespread consumption of caffeine during pregnancy alongside pregnancy-related changes in pharmacokinetics. I see the novelty and value of the work in its attempt to stitch together mechanistic evidence (reduced CYP1A2 activity, inter-individual metabolic differences) with clinical findings and in its separation of maternal outcomes (GDM, hypertension/preeclampsia) from neonatal outcomes (SGA/LBW, preterm birth). The recent literature update helps organize scattered signals and—once the messaging is refined—can meaningfully support nutrition counseling in pregnancy. The text has translational potential: it offers an opportunity to formulate clear “in-clinic” recommendations, provided that conclusions are explicitly tied to the type of exposure assessment, dose, and dietary context.

To strengthen this contribution, please distinguish sections based on self-reported intake from those based on biomarkers. A helpful complement would be a “take-home box” for clinicians: indicative daily intake thresholds (mg/day), differences between caffeine sources (coffee, tea, cocoa/chocolate, sugar-sweetened and energy drinks, decaf with residual caffeine), and guidance on trimester-specific considerations and timing relative to iron supplementation. It is also worth highlighting effect moderators (tobacco smoking, CYP1A2 polymorphisms, BMI, iron status, overall dietary pattern), which can help explain the heterogeneity of results. In the Supplement, I suggest adding a simple risk-of-bias assessment (e.g., low/moderate/high), a “direction of effect” column (↔/↑/↓), and a glossary of abbreviations so that the table is self-contained. Unifying terminology (e.g., preterm <37 weeks, SGA <10th percentile, LBW <2,500 g) and consistently reporting doses in mg/day within the narrative would also help. Finally, I recommend trimming repetitions between the abstract and conclusions and anchoring the take-home messages more tightly to the exposure type and dose ranges, with a clear indication of where the evidence is essentially neutral and where signals emerge at higher intakes and/or with co-exposures (e.g., sugar/taurine in energy drinks).

I would appreciate a point-by-point response to these issues, indicating where the corresponding changes have been made in the manuscript.

Best regards,

The reviewer.

Author Response

(The authors gave the same response as above.)

Round 2

Reviewer 3 Report

Comments and Suggestions for Authors

Dear Authors,

Thank you very much for your feedback. The answers are thoroughly satisfying.

Best regards,

The reviewer.